# A scoping review protocol of the lived experiences of doing a PhD in Africa

**Oluwatomilayo Omoya** [1] *, **Udeme Samuel Jacob**[2], **Olumide A. Odeyemi**[3], **Omowale A. Odeyemi**[4]

**1** College of Nursing and Health Science, Flinders University, Bedford, Australia, **2** Faculty of Education, Department of Educational Psychology, University of Johannesburg, Johannesburg, South Africa, **3** Research Division, University of Tasmania, Hobart, Australia, **4** Centre for Child & Adolescent Mental Health (CCAMH), University of Ibadan, Ibadan, Nigeria

* Oluwatomilayo.omoya@flinders.edu.au

## Abstract

### Objective

This scoping review aims to investigate the available literature on the lived experiences of doing a PhD in African Universities.

### Introduction

The continent of Africa still contributes a minimal amount of research towards global research outputs. The need for increased research capacity and outputs have been identified as priority for growth and development. There is a substantial need for evidence-based solutions that can alleviate some of these complexities. For example, challenges still exist in the disease burden faced, economic poverty and lack of infrastructure in various contexts.

### Methods and analysis

Multiple databases will be searched, including the EBSCO Host, Scopus, EMBASE, Cumulative Index to Nursing and Allied Health Literature (CINAHL), Medline (Ovid), and Google Scholar. The scoping review will be conducted using the Arksey & O'Malley (2005)'s six-step approach in conjunction with the Joanna Briggs Institute (JBI) methodology for scoping reviews. Studies that examined the perspective of PhD (Doctor of Philosophy) candidates, supervisors from Africa, and research studies focusing on the common barriers and facilitators concerning research in Africa will be included. Studies that explore the perspectives of other postgraduate cohorts will be excluded.

### Ethics and dissemination

Ethics application will not be required but findings will be disseminated through publications, conference presentation, policy, and relevant stakeholders.

### Review registration number

This study has been registered with the Open Science Forum (OSF).

**Data Availability Statement:** No datasets were generated or analysed during the current study. All relevant data from this study will be made available upon study completion.

**Funding:** The author(s) received no specific funding for this work.

**Competing interests:** The authors have declared that no competing interests exist.

## Introduction

Africa is the second largest continent, with approximately 16% of the global population. But contribute just 2 percent to the global expenditure on research development and output [1,2]. The reason for this is complex and challenging [3]. Studies have shown barriers such as lack of investment in research, a significant brain drain caused by massive emigration of skilled researchers to other continents,lack of funding, poor economic and scientific infrastructure all contributes to the complexity [2,3]. Africa has equally been disadvantaged as a subject of research that can be one-sided and biased views by non-African and mostly western perspectives [4]. Reports by the British Council and the German Academic Exchange Service (DAAD) have recognised that one of the important ways of facilitating development and advancement in Africa is to strengthen advanced research skills and training with efficient doctoral training [5,6]. Doctoral degrees are the highest academic qualifications received for demonstration of high research-level capability and original contribution to knowledge [7]. It is critical for knowledge production, and the pursuit to produce more PhD degree holders in Africa has increasingly become a priority [8]. African PhD graduates can better focus on problems that are relevant within the African context with the potential to provide feasible solutions. Within the last few decades, there have been several initiatives to reform doctoral training in Africa and increase research outputs [6]. However, low research productivity and lack of adequate staff with PhD qualifications in academia still plaque the system. There is a significant lack of evidence within the literature that explores the experience of PhD candidates in Africa. Therefore, a scoping review will be undertaken to determine the current and emerging evidence concerning the area of focus. The review will also determine the types of methods that are used in the available study to highlight gaps that are present in the literature. Moreover, the scoping review will determine whether a systematic review will be feasible. A preliminary search of MEDLINE, and JBI Evidence Synthesis was conducted and no current scoping reviews on the topic were identified. The objective of the scoping protocol is to provide a guide and map to assess the extent of the available literature on doing a PhD in Africa.

The use of a systematic review was considered but would not have satisfactorily provided answers to the aim of the research questions. The focus of the project is to perform a scoping review of the extent of the available literature on the topic of PhD experiences within the African context. One of the main reasons for this is a significant lack of available evidence. Therefore, the scoping review approach best fits as the method of inquiry because it allows an exploration of the depth and breadth of the available literature while mapping the evidence to inform future research.

### Aim/Objectives

The aim of the scoping review is to explore the available evidence concerning the lived experiences of doing a PhD in African universities with consideration to the barriers and facilitators that impact education and training. Guiding the scoping review aim are the following objectives that will:

1. determine the types of methods used in the identified studies, the knowledge of the breadth and extent of the available literature to determine gaps for future research

2. explore the definitions of terms and description of concepts used in the identified articles.

3. determine whether the study will be a precursor to inform a systemic review?

### Patient and public involvement

"No patient/public involved"

## Method

### Protocol design

The proposed scoping review will be conducted in accordance with the Arksey & O'Malley (2005)'s six-step approach in conjunction with the JBI methodology for scoping reviews [9,10]. These two approaches will define the review's objectives, methods, and dissemination process. The inclusion and exclusion criteria that will be employed, sources of evidence, data extraction and presentation. The JBI approach to conducting and reporting scoping reviews is consistent with the Preferred Reporting Items for Systematic reviews and Meta-Analyses-extension checklist for Scoping Reviews (PRISMA-ScR) (S1 and S2 Figs in S1 File) that will be employed in this protocol [10–14]. The scoping review title has been registered with the Open Science Forum: https://doi.org/10.17605/OSF.IO/AK7DY.

**Step 1: Developing the research question.** In developing the protocol review question, the "PCC" mnemonic will be used as a guide to ensure that the question was clear, and that the topic of the scoping review was efficiently addressed. The scoping review protocol was titled "A scoping review protocol to understand the lived experiences of doing a PhD in Africa" From the research question. The title of the protocol was structured to reflect the PCC mnemonic. For example, from the PCC mnemonic; P stands for the population, and in this protocol, these are the "PhD candidates". The C stands for a concept, which is the "lived experience of doing a PhD", and the C-context is "Africa" Guiding the scoping review are the broad questions listed below:

1. What are the barriers and facilitators that impact on the education and training of PhD candidates in Africa?

2. What are the roles of international collaborations in building research capacity and outputs?

3. What impact does research policies and agendas have on the quality of research and doctoral training?

4. What are factors driving the introduction and expansion of PhD provision in African universities?

5. What are the aspects that interfere with industry engagements, social issues, and private business?

**Step 2: Identifying relevant studies.** Multiple databases will be searched, including the EBSCO Host, Scopus, EMBASE, Cumulative Index to Nursing and Allied Health Literature (CINAHL), Medline (Ovid), and Google Scholar to identify relevant literature. The search method will usethe following keywords and phrases singly and then in combination: *"Experiences of doing a PhD"* OR *"Attitudes"* OR *"Lived Experience"* OR *"Perception"* AND *"PhD Candidate in west Africa"* OR *"Doctoral Candidate in Africa"* OR *"PhD Students in Sahara"* OR *"Doctoral Students in Nigeria"* AND *"West Africa"* OR *"South Africa"* OR *"Sahara"* OR *"Dark Continent"* OR *"East Africa" OR "North Africa"*. As recommended by the JBI review methods, a three-step search strategy will be employed. The first step will utilise the use of EBSCO Host, Scopus, EMBASE, CINAHL and MEDLINE databases. From this search, a list of keywords, and phraseswill be generated by analysing the title and abstract of the identified studies. The second step of the database search will be undertaken using the newly generated keywords across the other databases. Finally, a manual search of the reference list and bibliographies of the articles will be performed to identify other relevant studies.

**Step 3: Study selection.** The initial search of the literature will be performed with no initial inclusion and exclusion criteria applied. All identified studies will be imported into Covidence; a software where all the authors will have access to manage and streamline the studies obtained. The results obtained will then be screened by examining their titles and abstract. Articles will be assessed against the eligibility criteria. The inclusion criteria are research studies examining participants who have undertaken their PhD in Africa, studies that focus on the barriers and facilitators that have impact on doctoral training in Africa, research studies written in English language with a 20-year limit range. Studies will be excluded if they were based on the perspectives of postgraduate students who are not doing a PhD, research studies not written in English language, and participants who have done their PhD outside of Africa. A full text retrieval of studies will be obtained and further reviewed against the inclusion criteria. At this stage, two members of the research team will independently screen the articles and any disagreement are resolved by unanimous decision or by the third member of the research team. All articles will be imported into Covidence. The quality of the review will be ensured by using Covidence to search and remove duplicate articles. Evaluation of each article will be based on the inclusion and exclusion criteria. A flowchart of the review using PRISMA shows the detailed process of the initial search to data extraction in Fig 1 [12].

**Step 4: Charting the data/data extraction.** At this stage, data will be extracted from the articles by two or more members of the team and incorporated into a template data extraction instrument using JBI methodology guidance for scoping reviews. The template used will be modified from exiting tool (JBI template source of evidence details, characteristics, and results extraction instrument) anddraft data extraction will be modified as needed through the process. Any modification will be reported in the scoping review. This step will allow a visual representation of the major findings of each article and how they address the aims of the scoping review. A draft extraction form is provided in Table 1 below:

**Step 5: Collating, summarising, and reporting the results.** Analysis of evidence in a scoping review is confined to a descriptive level as the aim of the review is to map out the available studies that address the topic of interest. The obtained data will be descriptively mapped using a basic qualitative content analysis of data as opposed to a synthesis of result which is undertaken in systematic reviews. Data will be presented using parameters such as the number of publications found, types of studies, PhD experiences descriptors, and these will be reviewed according to the findings. Similar data sets will be grouped together and summarised with the aim of identifying gaps in the literature.

**Step 6: Acknowledgement/Consultation.** No consultations will be required for the scoping review and this protocol will not contribute towards a degree award.

**Ethics and dissemination.** This protocol will not require ethics approval because human/animals were not involved in the process. The results will however be disseminated through publications, conference presentation, relevant stakeholders, and policy.

## Strengths and limitations of this study

- The review will be one of the first study to examine the experience of PhD candidates in Africa.

- Exploring this issue has the potential to highlight the areas that impact on the education and training of PhD candidates.

- It would also inform higher institutions, supervisors, stakeholders and policy makers about the challenges and the needs for training doctoral students so that support and resources are directed to strengthen and increase efficiency.

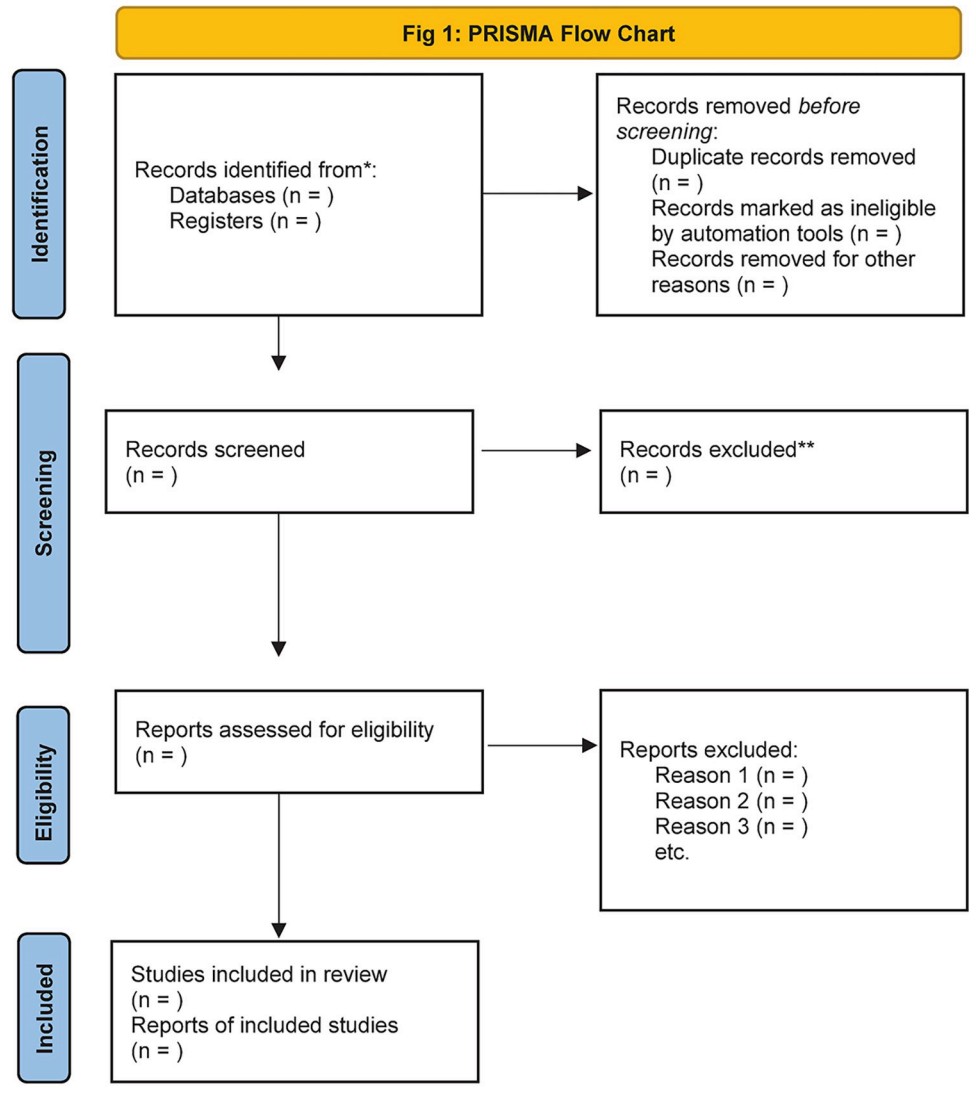

PRISMA 2020 V1 - Databases and Registers

**Fig 1. PRISMA flow chart.** The PRISMA flow chart is the Preferred Reporting Items for Systematic Reviews and Meta-Analyses. The results will initially be identified from the included databases. All the retrieved studies will then be removed based on the abstract and title. Records will then be screened against the eligibility criteria before they are finally included into the scoping review [12]. Moreover, the reference lists of articles will be searched manually. Additional studies will be identified by screening the reference lists of studies included in the electronic search [14].

**Table 1. Data extraction instrument.**

| |
|---|
| Author, Date and Location |
| Title of article |
| Aims/purpose |
| Sample Size and setting |
| Journal type |
| Methodology |
| Key Findings |

**Table 1.** The table is a modified version of the existing JBI data extraction guide. It will be used to guide data extraction from the retrieved studies.

- A limitation of the review lies in the potential for over representation of research articles from some African countries as opposed to others.

- Another limitation is the exclusion of articles that are not in English since the reviewers do not resource to interpret article in other language that is not English.

## Supporting information

**S1 File. S1 and S2 Figs.** These figures are a checklist used for reporting scoping protocol reviews and a draft of the search strategy.
(PDF)

## Author Contributions

**Conceptualization:** Oluwatomilayo Omoya, Olumide A. Odeyemi.

**Methodology:** Oluwatomilayo Omoya, Udeme Samuel Jacob, Olumide A. Odeyemi, Omowale A. Odeyemi.

**Supervision:** Oluwatomilayo Omoya.

**Validation:** Oluwatomilayo Omoya.

**Writing – original draft:** Oluwatomilayo Omoya.

**Writing – review & editing:** Oluwatomilayo Omoya, Udeme Samuel Jacob, Olumide A. Odeyemi, Omowale A. Odeyemi.

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
