## [Decision Letter · Decision Letter 0]

23 May 2023

PONE-D-23-01657A scoping review protocol to understand the lived experiences of doing a PhD in Africa.PLOS ONE

Dear Dr. Omoya,

Thank you for submitting your manuscript to PLOS ONE. After careful consideration, we feel that it has merit but does not fully meet PLOS ONE’s publication criteria as it currently stands. Therefore, we invite you to submit a revised version of the manuscript that addresses the points raised during the review process.

We look forward to receiving your revised manuscript.

Kind regards,

Olushayo Oluseun Olu

Academic Editor

PLOS ONE

3. Please include a new copy of Table 1 in your manuscript; the current table is difficult to read. Please follow the link for more information: https://blogs.plos.org/plos/2019/06/looking-good-tips-for-creating-your-plos-figures-graphics

Reviewers' comments:

Reviewer's Responses to Questions

**Comments to the Author**

1. Does the manuscript provide a valid rationale for the proposed study, with clearly identified and justified research questions?

Reviewer #1: Partly

Reviewer #2: Partly

2. Is the protocol technically sound and planned in a manner that will lead to a meaningful outcome and allow testing the stated hypotheses?

Reviewer #1: Partly

Reviewer #2: No

3. Is the methodology feasible and described in sufficient detail to allow the work to be replicable?

Reviewer #1: Yes

Reviewer #2: Yes

4. Have the authors described where all data underlying the findings will be made available when the study is complete?

Reviewer #1: Yes

Reviewer #2: No

5. Is the manuscript presented in an intelligible fashion and written in standard English?

Reviewer #1: Yes

Reviewer #2: No

6. Review Comments to the Author

You may also provide optional suggestions and comments to authors that they might find helpful in planning their study.

Reviewer #1: o In terms of the analysis the following observations are noted:

1. Line 94 – 100 is part of the introduction as a justification for the study method chosen.

2. The search strategy referred to as “Step 2 Identifying relevant studies” refers only to health related databases. For example, utilising MEDLINE and MeSH to generate “key words, scientific terms and medical subject headings” for use in other database searches. If the scoping review is limited to medical/health related PhD candidate experiences only, this needs to be stated clearly in the study title, research question and introduction section.

3. In the search strategy using the broad terms “experience ”or “attitude” singly is not efficient.

4. The search should consider using individual country names (54) not limited to “Africa”.

5. The Boolean operator approach should be cited as the method described for combining key words and synonyms.

6. Describe the process of how duplicates will be identified and removed

7. Describe how reference lists in identified articles will be further investigated.

8. As a scoping review the following objectives may also be considered:

a) The types of methods used in the identified studies and gaps

b) The definitions of terms and description of concepts used in the identified articles

c) Can the study be used as a precursor to inform a systemic review?

9. Under study limitations use of published literature only (not grey literature where several other types of evidence on the topic maybe available such as policy reports, national development plan reviews etc).

10. Also language limitations (Selecting for English only literature likely to excludes a large proportion of the literature).

Reviewer #2: Author need to reconcile the title with the main thrust in the article. It would seem that author wish to "document lived experiences" rather than "understand...lived experiences". Line 36: author needs to clarify if Africa has equally been disadvantaged as subject of research. Author need to explain what is "brain drain" (line 68). Line 78 - 79 clarify the decade in focus. There is lack of consistency between the introduction and the aim of study. The scope and boundaries of the scoping review needs to be better defined. Line 194 - should be supported with evidence in form of literature citation. 206 indicate that research was already conducted while mode of writing shows research "scoping" is to be done. Need to reconcile. Authors need to show better relationship between the PRISMA flow chart and the focus of study

7. PLOS authors have the option to publish the peer review history of their article (what does this mean?). If published, this will include your full peer review and any attached files.

Reviewer #1: **Yes: **Dr. Caroline Sarah Ryan

Reviewer #2: No

---

## [Author Response · Author response to Decision Letter 0]

31 Jul 2023

1. Line 94 – 100 is part of the introduction as a justification for the study method chosen. This Section has now been joined with the introduction so that it flows better. The method section has been moved to the protocol design as this best represent how the protocol will be undertaken. 

2. The search strategy referred to as “Step 2 Identifying relevant studies” refers only to health-related databases. For example, utilising MEDLINE and MeSH to generate “key words, scientific terms and medical subject headings” for use in other database searches. If the scoping review is limited to medical/health related PhD candidate experiences only, this needs to be stated clearly in the study title, research question and introduction section. Other databases have now been included and MeSH will not be used so that articles that capture experiences of participants from other areas will be included. 

3. In the search strategy using the broad terms “experience” or “attitude” singly is not efficient. Phrases have been added so that the search will be more efficient.

4. The search should consider using individual country names (54) not limited to “Africa”. This has now been addressed so that West, East, South and North Africa have been included in the search terms. Nigeria was specifically included because Nigeria has been reported to have a significant amount of PhD candidates compared to other African Countries

5. The Boolean operator approach should be cited as the method described for combining key words and synonyms. This has been addressed as well.

6. Describe the process of how duplicates will be identified and removed. All the articles will be uploaded into Covidence a software that is used to screen references and extract data. This software automatically removes any duplicates that are found within the collection of articles. This has been addressed in step 3: study selection. 

7. Describe how reference lists in identified articles will be further investigated. This has been addressed in step 2.

8. As a scoping review the following objectives may also be considered:

a) The types of methods used in the identified studies and gaps

b) The definitions of terms and description of concepts used in the identified articles

c) Can the study be used as a precursor to inform a systemic review? This have been included in the objectives and aims of the scoping review. 

9. Under study limitations use of published literature only (not grey literature where several other types of evidence on the topic maybe available such as policy reports, national development plan reviews etc).This has been removed so that all other relevant information will not be missed.

10. Also language limitations (Selecting for English only literature likely to excludes a large proportion of the literature). This has been addressed. 

Reviewer #2: Author need to reconcile the title with the main thrust in the article. It would seem that author wish to "document lived experiences" rather than "understand...lived experiences".The title has been changed so that it represents the aims of the review. 

 Line 36: author needs to clarify if Africa has equally been disadvantaged as subject of research. Author need to explain what is "brain drain" (line 68). Yes, this is an important point and has now been discussed. 

Line 78 – 79 clarify the decade in focus. There is lack of consistency between the introduction and the aim of study. The scope and boundaries of the scoping review needs to be better defined. This is from the 2000s, this has now been clarified. 

Line 194 - should be supported with evidence in form of literature citation. 206 indicate that research was already conducted while mode of writing shows research "scoping" is to be done. Need to reconcile. Authors need to show better relationship between the PRISMA flow chart and the focus of study. This statement has now been supported with research. This has been reworded to reflect what will be performed for the scoping review in the future.

---

## [Editor Report · Decision Letter 1]

15 Aug 2023

A scoping review protocol of the lived experiences of doing a PhD in Africa.

PONE-D-23-01657R1

Dear Dr. Omoya,

We’re pleased to inform you that your manuscript has been judged scientifically suitable for publication and will be formally accepted for publication once it meets all outstanding technical requirements.

Kind regards,

Olushayo Oluseun Olu

Academic Editor

PLOS ONE

Additional Editor Comments (optional):

I would recommend that the manuscript is copy-edited and proofread to correct the typographical and grammatical errors before final publication.
---

## [Editor Report · Acceptance letter]

24 Aug 2023

PONE-D-23-01657R1 

A scoping review protocol of the lived experiences of doing a PhD in Africa. 

Dear Dr. Omoya:

I'm pleased to inform you that your manuscript has been deemed suitable for publication in PLOS ONE. Congratulations! Your manuscript is now with our production department. 

Kind regards, 

on behalf of

Dr. Olushayo Oluseun Olu 

Academic Editor

PLOS ONE